# Towards Efficient Vision-Language Tuning: More Information Density, More Generalizability

## Abstract

With the advancement of large pre-trained vision-language models, effectively transferring the knowledge embedded within these foundational models to downstream tasks has become a pivotal topic, particularly in data-scarce environments. Recently, parameter-efficient fine-tuning approaches, especially prompt tuning, have garnered considerable attention. To better understand the nature of prompt tuning, we propose the concept of "Information Density" (ID) to indicate whether a matrix strongly belongs to certain feature spaces rather than being evenly distributed across various feature spaces. We suppose a higher ID with strong bias across some feature spaces naturally leads to excellent robustness and stability. Our research, inspired by the observation that generalizability is closely linked to the information density of the prompt embedding, introduces the Dense Information Prompt (DIP). DIP aims to enhance information density to improve generalization. Several alternative algorithms to increase ID are proposed and verified effective. With further help of proper initialization and regularization, comprehensive experiments substantiate the superiority of DIP. Notably, DIP surpasses the latest state-of-the-art methods by a substantial margin with an exceptionally small parameter count and no extra inference overhead. Across a range of tasks spanning 11 datasets, DIP improves the average downstream accuracy of classic prompt tuning by up to 5.76%.

## 1 Introduction

In recent years, vision-languages models Radford et al. (2021); Jia et al. (2021) have achieved tremendous success. Representative models like CLIP Radford et al. (2021) are first pre-trained on a huge number of text-image pairs on the web to align textual and visual features, and then can be tuned and used for various downstream tasks.

However, traditional fine-tuning is not a good choice to adapt vision-language models. Simply fine-tuning all the parameters can easily cause the model to overfit because the huge number of parameters bring redundant non-essential information. The huge training and storage cost is also an intractable problem. In the context of our study, we introduce the concept of "information density". Much like the rank of a matrix in linear algebra, which represents the maximum number of linearly independent rows or columns in the matrix, "information density" represents the maximum amount of essential and non-redundant information that the model can extract from the downstream task. Just as a matrix with a higher rank possesses more unique information, a model with high information density can acquire more essential and general information from the downstream task by fewer parameters, even with a smaller dataset. Our goal is increasing the information density and thus using the fewest but most essential parameters to finish generalization, without causing catastrophic forgetting or overfitting to the small dataset.

As the concept of information density can be functionally analogous to the rank of a matrix, we decided to use properties related to rank to quantify information density. Specifically, we take a full-rank matrix and decompose it using Singular Value Decomposition (SVD) to obtain a matrix of singular values, using the properties of these singular values to define and quantize information density. We found that this definition of information density is highly correlated with the model's

generalization performance (Spearman correlation coefficient > 0.9). Therefore, we propose DIP, aiming to enhance the model's generalization ability by increasing information density. Additionally, due to the increased information carried by each parameter unit, our approach can significantly reduce the required number of parameters.

In Section 3, we will show our finding about the strong correlation between generalization capability and information density of the prompt matrix in Fig. 1. Inspired by such observation, we propose Dense Information Prompt (DIP) for effective and efficient adaptation, well adapting models under an extremely small number of parameters where nobody has explored before. There are several good advantages of DIP:

- **Efficiency and Effectiveness** By operating on lightweight prompts, we can reach comparable or even better performance with state-of-the-art methods using very few parameters.
- **Simplicity** Replacing the classic prompt by DIP just needs to modify several lines of code.
- **Robustness** DIP is relatively capable of anti-disturbance. As in Tab. 3, DIP could maximally reserve its knowledge from domain shift.
- **Plug and Play** For any classic model, DIP only replaces the prompts, enabling us to plug DIP into most of the existing methods fruitfully.

In summary, we conclude our contributions as follows:

- We propose a new concept "information density", give its definition and well-quantize the concept. We further find the strong correlation between generalizability and information density in Fig. 1, and thus propose DIP to increase information density for better generalization capability.
- We design three effective implementations for DIP and prove that our motivation could be correctly verified through controlled experiments, *i.e.* improving ID significantly promotes generalizability.
- We propose a novel initialization technique and integrate a lightweight regularization module to further improve the performance of Dense Information Prompt tuning without introducing any extra parameters and inference cost.
- We conduct extensive experiments and show the fantastic effectiveness and efficiency of DIP. In base-to-new generalization, domain generalization, cross-dataset transfer and few-shot learning settings, DIP consistently reaches very competitive results, surpassing state-of-the-art tuning methods.

## 2 RELATED WORKS

### 2.1 VISION-LANGUAGE MODELS

Recently, large-scale vision-language models have shown very competitive performance in various tasks. Classic works Radford et al. (2021); Jia et al. (2021); Zhai et al. (2022); Yao et al. (2022); Yuan et al. (2021) learn the multi-modal representation by a self-supervised manner on a large amount of image-text pairs. The representative work CLIP Radford et al. (2021) is a milestone, which aligns the vision representation and language representation by contrastive learning and shows excellent performance. A well-trained vision-language model is a great treasure, which could largely facilitate the development of many fields. There have been successful applications of such strong models on few-shot recognition Zhou et al. (2022b;a), detection Rasheed et al. (2022); Maaz et al. (2022); Feng et al. (2022); Zang et al. (2022) and segmentation Li et al. (2022); Rao et al. (2022); Ding et al. (2022); Lüddecke & Ecker (2022). For video data, there are also works on video classification Qian et al. (2022) and video understanding Ju et al. (2022).

### 2.2 PROMPT TUNING

Prompt tuning is one of the most popular methods to tune models in downstream tasks with excellent efficiency. Originating from natural language processing, prompts are first introduced as a fixed template Schick & Schütze (2020), *e.g. a photo of a _*, which is hand-crafted and fixed. Later, a series of methods Li & Liang (2021); Lester et al. (2021); Liu et al. (2021b); Shin et al. (2020); Liu et al.

(2021a); Jiang et al. (2021) are proposed to make such prompts tunable and be optimized during adaptation. Prompt tuning could adaptively narrow the gap between pre-trained representations and downstream tasks, significantly facilitating the fine-tuning process. Representative prompt tuning methods would add tunable virtual tokens, *i.e.* prompts, along with the semantic tokens as inputs of the model. All of the tokens are processed together to get text embeddings first and then sent to the feature encoder. Witnessing the success of prompting language models, researchers design prompts Jia et al. (2022); Zhang et al. (2022) for visual models in a similar way. In vision-language field, there are several explorations as well. Bahng *et al.* Bahng et al. (2022) adopts prompt tuning merely on the image encoder. CoOp Zhou et al. (2022a) uses tunable text prompts to replace the fixed template in CLIP Radford et al. (2021). CoCoOp Zhou et al. (2022b) utilizes image feature to instruct the optimization of the tunable text prompts in CoOp. Khattak et al. (2023a); Lee et al. (2023) simultaneously optimize image and text prompts and establish extra connections between different modals. Khattak et al. (2023b); Yao et al. (2023); Bulat & Tzimiropoulos (2023); Zheng et al. (2023); Hao et al. (2024) integrate strong regularization modules or losses into prompt tuning to diminish the overfitting and catastrophic forgetting problem. For better downstream accuracy, researchers design more and more complicated methods, accompanied by inefficiency. To solve the problem, we propose Dense Information Prompt (DIP) to take the place of classic prompts, which can largely decrease the number of tunable parameters and further enhance the model's generalization ability. Notice that though becoming more complex, existing methods are still refined on a common fundamental basis, *i.e.* prompt tuning. Such a common basis guarantees that DIP could be easily and smoothly integrated into most of the off-the-shelf methods besides individually applied. Besides prompt-based methods, there are also many other works to acquire storage efficiency Hao et al. (2023a); Houlsby et al. (2019); Hu et al. (2021); Lian et al. (2022); Zhang et al. (2022); Chen et al. (2022a), inference efficiency Chen et al. (2022b); Wang et al. (2023a); Bolya et al. (2023); Hao et al. (2023b); Ding et al. (2021; 2019); Chen et al. (2023); Shen et al. (2024); Xiong et al. (2024) and data efficiency Wang et al. (2023b); Lyu et al. (2024) during downstream tuning.

## 3 RELATIONSHIP BETWEEN INFORMATION DENSITY AND GENERALIZABILITY

In this section, we will start from reviewing a classic prompt tuning pipeline CoOp Zhou et al. (2022a) on CLIP in Section 3.1, and propose a new concept "Information Density" and analyze its relationship with generalizability in Section 3.2.

### 3.1 A REVIEW OF PROMPT TUNING FOR CLIP

CLIP consists of a text encoder $\mathcal{L}$ and an image encoder $\mathcal{V}$. Typically, $\mathcal{L}$ is a language transformer, while $\mathcal{V}$ can be a convolutional neural network or a vision transformer. In this paper, we follows Zhou et al. (2022a;b) to use a ViT-B/16 Dosovitskiy et al. (2020) as the image encoder $\mathcal{V}$ unless specifically mentioned. We start by making a review of how to prompt a CLIP for prediction in the following paragraphs.

**Text Encoder** Suppose there are $M$ layers in the text encoder. For $k$-th layer $\mathcal{L}_k$, the inputs are a series of prompt tokens $P_{k-1}^l$ and a *[CLS]* token $c_{k-1}^l$, and the outputs are $P_k^l$ and $c_k^l$. The inputs of the first layer $P_0^l$ and $c_0^l$ are exactly the word embeddings of the prompts along with the label, *e.g.* "*A photo of a [CLS]*" or just some randomly initialized vectors. Formally, we have $P_k^l \in \mathbb{R}^{n^l \times d^l}$ and $c_k^l \in \mathbb{R}^{d^l}$, where $n^l$ denotes the text prompts' length and $d^l$ denotes the dimension of word embedding. $\forall 1 \leq k \leq M$, we have $[P_k^l, c_k^l] = \mathcal{L}_k([P_{k-1}^l, c_{k-1}^l])$.

The output feature of the text encoder $f^l \in \mathbb{R}^{d^v}$, where $d^v$ is the dimension of the visual feature space, is generated by projecting the *[CLS]* token of the last layer to the visual space by a linear transformation, *i.e.* $f^l = \text{Proj}(c_M^l)$.

**Image Encoder** Suppose there are $N$ layers in the image encoder. For $k$-th layer $\mathcal{V}_k$, the inputs are image patch tokens $I_{k-1}$, a classification token $c_{k-1}^v$ and prompt tokens $P_{k-1}^v$, and the outputs are $I_k$, $c_k^v$ and $P_k^v$. The inputs of the first layer $I_0$ and $c_0^v$ are exactly the patch embeddings of the image and pre-trained class token. $P_0^v$ is randomly initialized in general. Formally, we have $I_k \in \mathbb{R}^{p \times d^v}$, $c_k^v \in \mathbb{R}^{d^v}$ and $P_k^v \in \mathbb{R}^{n^v \times d^v}$, where $p$ denotes the number of image patches and $d^v$ denotes the

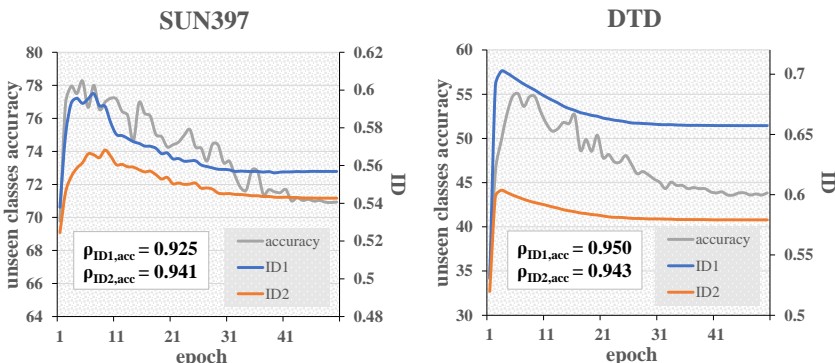

Figure 1: Relationship between generalizability represented by the test accuracy on unseen classes during training and Information Density (ID). When generalizability increases, ID also increases. The Spearman correlation coefficient $\rho$ between generalizability and ID1/ID2 is very high, *i.e.* $\geq$ 0.9.

dimension of visual embedding. $\forall 1 \leq k \leq N$, $[P_k^v, c_k^v, I_k] = \mathcal{V}_k([P_{k-1}^v, c_{k-1}^v, I_{k-1}])$. The output feature of the image encoder is $f^v = c_N^v$.

**Prediction** CLIP can be used for image classification. Suppose there are $C$ classes, and $\{f_c^l\}_{c=1}^{C}$ are the corresponding text features. Label $y$'s probability is $p(y|f^v) = \frac{\exp(\text{sim}(f^v, f_y^l)/\tau)}{\sum_{c=1}^{C} \exp(\text{sim}(f^v, f_c^l)/\tau)}$ where $\text{sim}(\cdot, \cdot)$ denotes cosine similarity function and $\tau$ is temperature. The final prediction is $\hat{z} = \underset{1 \leq y \leq C}{\arg\max}(p(y|f^v))$.

It is worth noting that some researchers adopt a deeper manner Jia et al. (2022); Khattak et al. (2023a) to organize the prompts. They directly add and tune the prompt in each layer in the feature encoder, instead of inheriting the output prompt calculated by the last encoder, *i.e.* a forward pass becomes $[\_, c_k^l] = \mathcal{L}_k([P_{k-1}^l, c_{k-1}^l])$ and $[\_, c_k^v, I_k] = \mathcal{V}_k([P_{k-1}^v, c_{k-1}^v, I_{k-1}])$. Each $P^l/P^v$ contains tunable parameters.

### 3.2 INFORMATION DENSITY IN PROMPT TUNING

Here, we first provide precise definitions for "information density" to clearly convey our motivations. For typical parameter matrix like prompts $P \in \mathbb{R}^{n \times d}$(assume $n < d$), we can always rewrite such matrix into a combination of several orthogonal bases with different weights by singular value decomposition (SVD). Formally, $P = U\Sigma V^T = \sum_{i=1}^{d} \sigma_i u_i v_i^T$. Each $u_i v_i^T$ can span a unique feature space, and $P$ is a linear combination of these features. Typically, $\{\sigma_i\}_{i=1}^{n}$ are arranged in descending order.

To help readers better understand the concept of information density, let's draw an analogy using images from nature. A real image always has one or a few very prominent features and almost never contains an even mix of various odd features. In an extreme case, if an image truly exhibits isotropic characteristics, it would simply mean that its content is almost entirely noise and lacks clear meaning. Reflecting this back to the matrix decomposition we discussed earlier, a good image tends to have several significantly larger singular values. The features of the image can largely be expressed by the feature space behind these prominent singular values. Therefore, from the matrix decomposition expression, the differences among the singular values $\{\sigma_i\}_{i=1}^{n}$ are significant, indicating that the information is concentrated in a few feature spaces. In other words, the information density is higher. Thus, to quantize such property, we define $k$-th order "Information Density (ID)" as follows: $\text{ID}k = \frac{\sum_{i=1}^{k} \sigma_i}{\sum_{i=1}^{n} \sigma_i}$. In other words, $\text{ID}k$ is the proportion of the largest $k$ singular values among all the singular values. Greater information density represents more robust and stable intrinsic features, meaning they are less likely to be affected by external disturbances and have stronger anti-interference capabilities.

Returning to our initial discussion, a core contribution of this paper is the hypothesis and verification that the parameter matrices, like prompts, follow the same ID-related laws during fine-tuning

in downstream CLIP models. As is well-known, in transfer learning, the transfer of knowledge in a model depends on the updating of parameters. For CLIP models, the optimal solution is prompt tuning Zhou et al. (2022a;b); Khattak et al. (2023a); Zhu et al. (2023). We first hypothesize that the information density of the prompt matrix also represents its robustness and the strength of its intrinsic features. Therefore, greater information density should theoretically result in better generalizability throughout the prompt tuning process.

To verify this hypothesis, we conducted an experiment using a classic method, CoOp Zhou et al. (2022a). During training, we masked half of the classes, using data from only the other half for training, and performed singular value decomposition on the prompt matrices during the process. As shown in Fig. 1, for better visualization, ID1 is scaled up to 2x to be put under the same right axis with ID2. Unseen classes accuracy is improved in the first few iterations, but it starts dropping later, indicating overfitting and catastrophic forgetting. Importantly, the fluctuation trend of unseen classes accuracy is highly consistent with ID. We compute the Spearman correlation coefficient between unseen classes accuracy and ID1/ID2 in Fig. 1. Clearly, the first-order and second-order information densities of CoOp on the SUN397 Xiao et al. (2010) and DTD Cimpoi et al. (2014) datasets exhibit a very strong correlation with the accuracy on unseen classes (*i.e.*, generalizability), with Spearman correlation coefficients greater than 0.9. This demonstrates that our hypothesis is correct.

In the following Section 4, we will show how we can leverage such correlation between generalizability and ID to boost CLIP's downstream performance.

## 4 METHODOLOGY

### 4.1 DENSE INFORMATION PROMPT

#### 4.1.1 ALGORITHMS FOR INCREASING INFORMATION DENSITY

Motivated by the observation in Section 3, we propose several algorithms to increase information density to enhance CLIP's generalizability.

**1. Direct Optimization (DO):** Do SVD and directly optimize information density as a training objective. Specifically, we add corresponding $\text{ID}k = \frac{\sum_{i=1}^{k} \sigma_i}{\sum_{i=1}^{n} \sigma_i}$ to the loss item after decomposing prompt matrix $P$ by $P = U\Sigma V^T$. Such a decomposition is done before each iteration starts. In other words, the parameter matrices we actually update are $U, \Sigma$ and $V$. Formally, suppose the original cross-entropy loss is $\mathcal{L}_{CE}$ and now we aim to maximize $\text{ID}k$, the new training loss is $\mathcal{L}_{\mathcal{DO}} = \mathcal{L}_{CE} - \lambda_{DO}\text{ID}k$, where $\lambda_{DO}$ is a positive hyper-parameter. After each training iteration $t$, a composition is required to obtain $P^{(t+1)} \leftarrow U^{(t)}\Sigma^{(t)}V^{(t)T}$ to restrict the freedom of accumulated parameter updates and keep its physics, while the tunable parameters $U^{(t+1)}, \Sigma^{(t+1)}$ and $V^{(t+1)}$ are got by decomposition of $P^{(t+1)} = U^{(t+1)}\Sigma^{(t+1)}V^{(t+1)T}$. As a result, the gradient of $U^{(t)}$ and $V^{(t)}$ are $\frac{\partial L_{DO}^{(t)}}{\partial U^{(t)}} = \frac{\partial L_{CE}^{(t)}}{\partial U^{(t)}}$ and $\frac{\partial L_{DO}^{(t)}}{\partial V^{(t)}} = \frac{\partial L_{CE}^{(t)}}{\partial V^{(t)}}$ separately. For each $\sigma_j^{(t)}(1 \leq j \leq n)$ in $\Sigma^{(t)}$, $\frac{\partial L_{DO}^{(t)}}{\partial \sigma_j^{(t)}} =$

$\frac{\partial L_{CE}^{(t)}}{\partial \sigma_j^{(t)}} - \lambda_{DO}\frac{\partial \text{ID}k^{(t)}}{\partial \sigma_j^{(t)}} = \frac{\partial L_{CE}^{(t)}}{\partial \sigma_j^{(t)}} - \lambda_{DO}\frac{\partial \frac{\sum_{i=1}^{k} \sigma_i^{(t)}}{\sum_{i=1}^{n} \sigma_i^{(t)}}}{\partial \sigma_j^{(t)}} = \frac{\partial L_{CE}^{(t)}}{\partial \sigma_j^{(t)}} - \lambda_{DO}\frac{I(j \leq k)(\sum_{i=1}^{n} \sigma_i^{(t)}) - \sum_{i=1}^{k} \sigma_i^{(t)}}{(\sum_{i=1}^{n} \sigma_i^{(t)})^2}$, where $I(*)$ is an indicator function.

**2. Positional Penalty (PP):** Do SVD and apply positional penalty on the singular values. Specifically, we add a kind of position-related regularization term for singular values to the training objective. Formally, suppose the original cross entropy loss is $\mathcal{L}_{CE}$ and now we aim to maximize $\text{ID}k$, the new training loss is $\mathcal{L}_{PP} = \mathcal{L}_{CE} + \lambda_{PP}\sum_{i=n-k+1}^{n} i\sigma_i^{(t)}$. Similar with DO, we do decomposition before each iteration and do composition after each iteration to restrict the freedom of accumulated parameter updates and keep its physics. As a result, the gradient of $U^{(t)}$ and $V^{(t)}$ are $\frac{\partial L_{PP}^{(t)}}{\partial U^{(t)}} = \frac{\partial L_{CE}^{(t)}}{\partial U^{(t)}}$ and $\frac{\partial L_{PP}^{(t)}}{\partial V^{(t)}} = \frac{\partial L_{CE}^{(t)}}{\partial V^{(t)}}$ separately. For each $\sigma_j^{(t)}(1 \leq j \leq n)$ in $\Sigma^{(t)}$, $\frac{\partial L_{PP}^{(t)}}{\partial \sigma_j^{(t)}} == \frac{\partial L_{CE}^{(t)}}{\partial \sigma_j^{(t)}} - \lambda_{PP}\frac{\partial \text{ID}k^{(t)}}{\partial \sigma_j^{(t)}}$.

**3. Low-Rank Approximation (LRA):** Approximate the original matrix by the product of two small matrices. Formally, given a typical learnable prompt $P_{DIP} \in \mathbb{R}^{n \times d}$, we create two low-

Table 1: A quick check for different implementations of DIP without initialization and regularization. All the proposed algorithms can improve generalizability, well verifying our former observation and motivation.

|         | base  | new   | H     |
|---------|-------|-------|-------|
| w/o any | 82.69 | 63.22 | 71.66 |
| DIP-DO  | 81.21 | 67.58 | 73.77 |
| DIP-PP  | 80.39 | 69.85 | 74.75 |
| DIP-LRA | 79.91 | 71.48 | **75.46** |

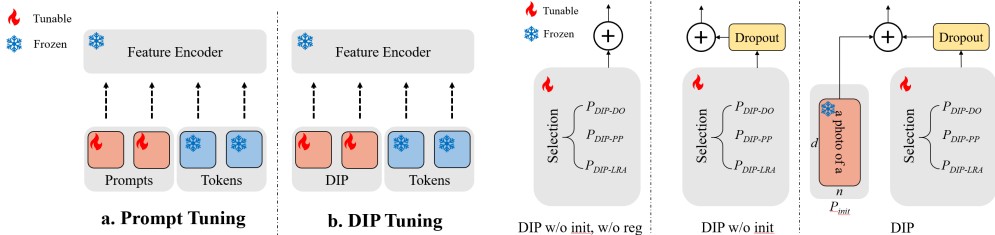

(a) Overview: DIP tuning and prompt tuning.     (b) DIP enjoys initialization and regularization.

Figure 2: To switch from classic prompt tuning and to DIP tuning, just replace the ordinary prompts with DIPs. For the prompts with special initialization, *e.g.* a hand-crafted template *"a photo of a"* for the text prompts, we introduce a concurrent normal prompt branch along with the proposed low-rank prompts. By turning off the gradient of the newly added branch, we start training from a promising initial point, and the total number of stored parameters will not increase as well. The Dropout layer could effectively regularize the update of DIP prompts and alleviate overfitting and catastrophic forgetting. Dropout is a lightweight non-parametric layer and turns out to be an Identity layer in inference, resulting in negligible cost.

dimensional matrices and use their product as an approximated equivalent prompt. Suppose we want to maximize ID$k$ ($1 \leq k \leq n$), we first randomly initialize $P_A \in \mathbb{R}^{n \times k}$ and $P_B \in \mathbb{R}^{k \times d}$. The low-rank approximated prompt $P_{DIP} = P_A P_B$ is with the same shape with the original one, and thus can participate in the training as usual. Parameters in $P_A$ and $P_B$ are updated through the gradients given by a normal loss $L_{LRA} = L_{CE}$.

In Tab. 1, we show that all of the three proposed algorithms to improve ID$k$ can promote generalization indeed, compared with baseline prompt tuning. In experiments, we select the best implementation from DIP-DO, DIP-PP and DIP-LRA according to the pre-experimental results on validation set.

### 4.1.2 INITIALIZATION

In the field of tuning vision-language models, existing works have confirmed that the initialization method of prompts is quite important. For example, Zhou et al. (2022b;a) adopt a hand-crafted template as the initial point of the text prompts, and Lee et al. (2023) copies the parameters in the text or image class token to initialize the prompts of the corresponding branch. If we just do a random initialization, the overall performance of such existing methods would drop severely. See Section 5.2 for more details. In other words, it would be helpful if we could take advantage of a good initial point.

The problem lies in the fact that directly applying dense information losses or structures on artificially designed initialization $P_{init}$ is not a good idea. Violet jitter might cause sever performance drop for DIP-DO and DIP-PP. As for DIP-LRA, since $k < min(n,d)$, it is impossible to directly initialize $P_A$ and $P_B$ by a given $P_{init}$.

To solve such a problem, we add a concurrent branch of normal frozen prompts $P_{init} \in \mathbb{R}^{n \times d}$ along with the proposed DIP as shown in Fig. 2b. For DIP-DO and DIP-PP, an additional random initialized prompt branch is used for update. For DIP-LRA, we randomly sample $P_A/P_B$ from a Gaussian distribution in which $\mu = 0$ and $\sigma \to 0$. A small $\sigma$ here could avoid constant initialization and enrich the update paths. Notably, $P_{init}$ is kept frozen during the whole adaptation.

Table 2: Comparisons with latest methods in base-to-new generalization. H: harmonic mean Xian et al. (2017). DIP can be fruitfully integrated into most prompt tuning methods, which are the mainstream research methods in this area. Integrating DIP into various state-of-the-arts outperforms the original baseline significantly on new and harmonic mean accuracy, and sometimes on base accuracy as well, showing great superiority and compatibility of our proposed method.

| Method | Base | New | H |
|---|---|---|---|
| **Non-prompt tuning** | | | |
| CLIP Radford et al. (2021) | 69.34 | 74.22 | 71.70 |
| Adapter Gao et al. (2021) | 82.62 | 70.97 | 76.35 |
| LoRA Hu et al. (2021) | **84.30** | 67.33 | 74.86 |
| **Prompt tuning** | | | |
| CoOp Zhou et al. (2022a) | **82.69** | 63.22 | 71.66 |
| DIP+CoOp | 80.32 | **74.73** | **77.42** |
| CoCoOp Zhou et al. (2022b) | 80.47 | 71.69 | 75.83 |
| DIP+CoCoOp | **80.62** | **73.68** | **77.00** |
| ProGrad Zhu et al. (2023) | **82.79** | 68.55 | 75.00 |
| DIP+ProGrad | 81.24 | **73.14** | **76.97** |
| MaPLe Khattak et al. (2023a) | 82.28 | 75.14 | 78.55 |
| DIP+MaPLe | **83.17** | **75.43** | **79.11** |
| DePT Zhang et al. (2024) | 85.15 | 76.06 | 80.35 |
| DIP+DePT | **85.18** | **76.66** | **80.70** |

### 4.1.3 REGULARIZATION

As discussed in Section 2, existing works have shown that proper regularization would significantly improve the generalization ability. Therefore, to alleviate overfitting and catastrophic forgetting, we put a Dropout layer with drop ratio $p$ after the DIP branch as displayed in Fig. 2b.

Therefore, the input prompt of the feature encoder is $P = P_{fr} + \text{Dropout}(P_{lr}, p)$. Finally, we have $P = P_{init} + \text{Dropout}(P_A P_B, p)$.

## 4.2 EFFICIENCY ANALYSIS

The whole fine-tuning process of a vision-language model can be divided into three parts: training, storage and inference. We separately analyze the efficiency of DIP in each part here.

**Training** In the training phase, all of DIP-DO, DIP-PP, DIP-LRA lead to slight training cost increase which is negligible. See Tab. 12 for more details.

**Storage** After training, DIP-DO and DIP-PP save the prompt parameters with the same size, *i.e.* $nd$, as the original prompt onto disk. DIP-LRA merely saves $k(n + d)$ parameters, less than $nd$ of the original prompt tuning.

**Inference** Before inference, we first load $P_{init}$ and $P_{DIP}$ from disk to memory. Noticing that Dropout is exactly an identity layer in the inference mode, we could pre-calculate the equivalent $P$ by $P = P_{init} + P_{DIP}$ and just keep $P$ in the memory. For inference, we directly use $P$ as the input prompts, and thus the inference cost is the same as classic prompt tuning. Some existing methods add complex bridges between the isolated parameters to earn extra improvements, *e.g.* CoCoOp Zhou et al. (2022b). There ain't no such thing as a free lunch. They would face slower speed and huge memory occupation in inference time.

## 5 EXPERIMENTS

To verify the effectiveness of the proposed method, we evaluate our method and make comparisons with the latest state-of-the-art methods in terms of the following settings in a wide range: base-to-new generalization, domain generalization, cross-dataset transfer and few-shot learning.

For more experimental details, please refer to the Appendix.

Table 3: Comparisons with latest methods in domain generalization after tuned on ImageNet. DIP integrated baselines shows excellent robustness when dealing with domain shift.

| | Source | | Target | | | |
|---|---|---|---|---|---|---|
| | ImageNet | IN-V2 | IN-S | IN-A | IN-R | Average |
| **Non-prompt tuning** | | | | | | |
| CLIP | 66.73 | 60.83 | 46.15 | 47.77 | 73.96 | 57.18 |
| Adapter | 69.33 | 62.53 | 47.67 | 49.17 | 75.42 | 58.70 |
| LoRA | 70.30 | 62.37 | 42.43 | 38.40 | 68.97 | 53.04 |
| **Prompt tuning** | | | | | | |
| CoOp | **71.51** | 64.20 | 47.99 | 49.71 | 75.21 | 59.28 |
| DIP+CoOp | 70.80 | 63.95 | **49.07** | **50.97** | **77.19** | **60.30** |
| CoCoOp | 71.02 | 64.07 | 48.75 | 50.63 | 76.18 | 59.91 |
| DIP+CoCoOp | **71.10** | **64.27** | **49.13** | **50.73** | **77.07** | **60.30** |
| ProGrad | **72.24** | **64.73** | 47.61 | 49.39 | 74.58 | 59.07 |
| DIP+ProGrad | 71.39 | 64.36 | **48.56** | **50.10** | **76.39** | **59.85** |

Table 4: Results in the cross-dataset transfer setting. DIP+CoOp gives the highest accuracy on 6 of 10 datasets, and slightly outperforms CoCoOp on average. Such result well demonstrates that DIP could maximally extract general and data-agnostic knowledge from given images.

| | Source | Target | | | | | | | | | | Average |
|---|---|---|---|---|---|---|---|---|---|---|---|---|
| | ImageNet | Caltech101 | Pets | Cars | Flowers | Food101 | Aircraft | Sun397 | DTD | EuroSAT | UCF101 | |
| CoOp | 71.51 | 93.70 | 89.14 | 64.51 | 68.71 | 85.30 | 18.47 | 64.15 | 41.92 | 46.39 | 66.55 | 63.88 |
| CoCoOp | 71.02 | **94.43** | 90.14 | 65.32 | **71.88** | 86.06 | 22.94 | 67.36 | 45.73 | 45.37 | 68.21 | 65.74 |
| Adapter | 69.33 | 93.43 | 88.87 | 64.40 | 70.27 | 85.63 | **24.67** | 65.80 | 44.90 | **47.70** | 66.00 | 65.17 |
| DIP+CoOp | 70.57 | 94.20 | **90.50** | **67.17** | 71.27 | **86.07** | 23.83 | **67.60** | **46.73** | 42.10 | **68.93** | **65.84** |

## 5.1 MAIN RESULTS

### 5.1.1 BASE-TO-NEW GENERALIZATION

The average results over 11 datasets are shown in Tab. 2. Overall, DIP can be fruitfully integrated into various state-of-the-art prompt tuning methods and further improve performance by a clear margin. The superiority mainly relies on the improvement of new classes. In other words, DIP largely improves the generalization ability of CLIP, which verifies our earlier observation and motivation. In particular, compared with latest DePTZhang et al. (2024), DIP gets 0.35% H accuracy gain with no extra inference cost.

### 5.1.2 DOMAIN GENERALIZATION

Then we follow CoCoOp Zhou et al. (2022b) to use ImageNet, ImageNet-A, ImageNet-R, ImageNet-v2, and ImageNet-S to run domain generalization experiments to verify the robustness of DIP. Shown in Tab. 3, on target datasets, DIP leads to better average accuracy compared with the latest methods, largely outperforming state-of-the-art baselines with significantly better resistance against domain shift.

### 5.1.3 CROSS-DATASET TRANSFER

Finally, we follow CoCoOp Zhou et al. (2022b) to conduct cross-dataset transfer evaluation. Results are shown in Tab. 4. Concentrating too much on the current dataset will absolutely cause overfitting and catastrophic forgetting problems, and finally lead to a severe drop in the performance on those unseen datasets. In this setting, DIP wins on 6 of 10 datasets and its average accuracy is also slightly better than the best competitor CoCoOp. Such result well demonstrates that DIP could maximally extract general and data-agnostic knowledge from given images compared with other prompt-based methods. Considering the huge difference in the parameter numbers, we could summarize that DIP is still the better choice.

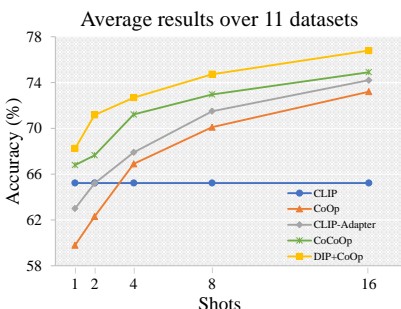

Figure 3: Few-shot learning Results.

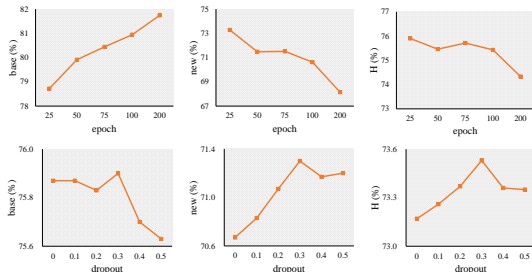

Figure 4: Top: Effect of training epochs. Bottom: Effect of dropout ratios.

Table 5: Ablation study on base-to-new generalization setting

|  | Dense Information | Initialization | Regularization | base | new | H |
|---|---|---|---|---|---|---|
| CoOp | - | - | - | **82.69** | 63.22 | 71.66 |
| DIP+CoOp | ✓ | - | - | 79.91 | 71.48 | 75.46 |
|  | ✓ | ✓ | - | 79.42 | 73.21 | 76.19 |
|  | ✓ | ✓ | ✓ | 79.70 | **73.59** | **76.53** |

### 5.1.4 FEW-SHOT LEARNING

In this paragraph, we will show the experiment results of DIP in the few-shot learning setting. This setting is originated from CoOp. Seen from Fig. 3, DIP consistently outperforms zero-shot CLIP, CoOp, and CLIP-Adapter across all the shot numbers. Such results demonstrate the superiority of DIP in adaptation ability when there are few samples in downstream tasks.

Overall, in base-to-new generalization, domain generalization, cross-dataset transfer and few-shot learning, DIP can be fruitfully integrated into existing methods and consistently reaches state-of-the-art performance while enjoying extremely high parameter efficiency.

## 5.2 ANALYSIS

### 5.2.1 ABLATION STUDY

We first show the impact of components of DIP step by step.

We start from transforming the text encoder. We first replace the tunable prompt tokens with our decomposed small prompt matrices, denoted as "Decomposition", for the CoOp method. After such replacement, the average Harmonic accuracy over 11 datasets directly improved from 71.66% to 75.46%. The accuracy on the base classes decreases by 2.78% and on the new classes increases by 8.26%. Although the adaptation ability slightly drops, the generalization ability raises quite a lot. This phenomenon proves that our dense information design is fairly beneficial for the model's generalization ability once again. Then we integrate the special initialization into CoOp. The harmonic mean accuracy improves by 1.27%. Finally, we add a lightweight regularization layer, Dropout. It helps us alleviate overfitting and catastrophic forgetting, resulting in further improvement.

### 5.2.2 ADDITIONAL EXPERIMENTAL RESULTS

**Effect of different training epochs** In this paragraph, we investigate that how the total training epoch could influence the adaptation result. Shown in Fig. 4, we run experiments for the given epochs separately. As the training epoch increases, the accuracy on base classes continues decreasing while the accuracy on new classes continues increasing. It is reasonable because as the training continues, the model has a higher risk of forgetting its original knowledge and overfitting.

**Effect of different dropout ratios** In this paragraph, we will show the influence of different dropout ratios in DIP on ImageNet. Seen from Fig. 4, as the dropout ratio increases, the base accuracy starts decreasing while the new accuracy starts increasing mostly. The harmonic mean first increases and then decreases. Dropout iss a kind of regularization, only proper regularization can help avoid overfitting and catastrophic forgetting.

Table 6: Results on RN-50 encoded CLIP.

|  | base | new | H |
|---|---|---|---|
| CoOp | **77.16** | 61.01 | 68.14 |
| ProGrad | 73.29 | **65.96** | 69.06 |
| DIP+CoOp | 75.22 | 64.98 | **69.73** |

Table 7: Effect of ID order on ImageNet.

| ID order | base | new | H |
|---|---|---|---|
| 1 | 75.87 | **70.67** | 73.17 |
| 2 | 76.03 | 70.13 | 72.96 |
| 3 | **76.20** | 70.40 | **73.19** |

Table 8: Results of adding DIP to image prompts on ImageNet.

|  | base | new | H |
|---|---|---|---|
| Text DIP | **75.87** | **70.67** | **73.17** |
| Image DIP | 74.57 | 69.40 | 71.89 |

**CLIP with convolutional image encoder** In this paragraph, we show the results of DIP on CLIP with convolutional image encoder ResNet-50 He et al. (2016), rather than the default ViT-B/16 Dosovitskiy et al. (2020). Seen from Tab. 6, compared with baseline CoOp, DIP still largely improves the new accuracy and the harmonic mean accuracy over 11 datasets, while the base accuracy slightly drops. Compared with the latest method ProGrad, DIP shows clear superiority on base accuracy and the harmonic mean accuracy.

**Effect of different ID order** In this paragraph, we will show the influence of different ID orders in DIP. Seen from Tab. 7, roughly, a larger order triggers higher base accuracy, and lower new accuracy as we expected before. In summary, increasing or decreasing ID order is not necessarily able to improve the average accuracy. It depends.

**Results for using DIP on the image prompts** There are several works Jia et al. (2022); Khattak et al. (2023a) indicating the effectiveness of the image prompts. Therefore, in this subsection, we will explore the results of applying DIP to image prompts. Seen from Tab. 8, using DIP on the image side also reaches high accuracy. However, the base, new, and average accuracy of image DIP is not as good as those of text DIP. Such experiment result tells us to use DIP on the text side instead of the image side when we aim to reach high accuracy with extremely few parameters.

## 6 CONCLUSION

With the development of huge vision-language models, how to effectively and efficiently adapt such huge models to downstream tasks becomes a challenging problem. Much effort has been made to leverage the potential of prompt tuning in adapting vision-language models. However, existing methods suffer from inefficiency. To reach extremely efficient generalization, we propose Dense Information Prompt (DIP) based on the observation about the strong correlation between Information Density (ID) and generalizability. Moreover, we propose a novel initialization method and a lightweight regularization module to further improve the dense information design without adding any extra inference cost. Besides efficiency and effectiveness, DIP has many valuable advantages such as simplicity and robustness. Extensive experiments and analyses adequately show the superiority of DIP.

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

## A  DATASETS

Following previous work Zhou et al. (2022a;b), we leverage 11 image recognition datasets to verify the effectiveness of the proposed method for both the base-to-new generalization task. These datasets include two datasets for the generic object classification, *i.e.*, ImageNet Deng et al. (2009) and Caltech101 Fei-Fei et al. (2004), five datasets for the fine-grained classification, *i.e.*, Oxford-Pets Parkhi et al. (2012), StanfordCars Krause et al. (2013), Flowers102 Nilsback & Zisserman (2008), Food101 Bossard et al. (2014) and FGVCAircraft Maji et al. (2013), one dataset for the scene recognition, *i.e.*, SUN397 Xiao et al. (2010), one dataset for the action recognition, *i.e.*, UCF101 Soomro et al. (2012), one dataset for the texture classification, *i.e.*, DTD Cimpoi et al. (2014), and one dataset for the satellite imagery recognition, *i.e.*, EuroSAT Helber et al. (2019). Following previous works Zhou et al. (2022b;a), for each dataset, we split its classes equally into two non-overlapping groups, *i.e.*, one as base classes and the other as new classes. We train all models on the base classes and perform a base/new evaluation on the base/new classes.

For the domain generalization task, we utilize ImageNet-A Hendrycks et al. (2021b), ImageNet-R Hendrycks et al. (2021a), ImageNetv2 Recht et al. (2019) and ImageNet-S Wang et al. (2019) to verify the robustness of the model. In this setting, we need to first train the model using ImageNet, and then directly use images from other four datasets to do inference.

For the cross-dataset transfer task, the datasets are the same as those of the base-to-new generalization task. Similar to domain generalization, the model will be first trained on ImageNet and then do inference on the other 10 different datasets.

For the few-shot learning task, the datasets are the same as those of the base-to-new generalization task. The model will be trained and evaluated with 1, 2, 4, 8 and 16 shots separately.

The dataset splitting is exactly the same as previous works Zhou et al. (2022a;b). We report the averaged model performance over three runs with different random seeds for fair comparisons.

## B  TRAINING DETAILS

Following previous work Zhou et al. (2022b), we employ ViT-B/16 as the image encoder in the CLIP. Each training image is resized to $224 \times 224$ before being fed into the image encoder. Some common data augmentation strategies, *e.g.*, random crop and random flip, are used to enhance the model performance, following Zhou et al. (2022b). During training, we set the batch size as 32. We employ the stochastic gradient descent algorithm (SGD) to optimize the learnable parameters. As Zhou et al. (2022a), we utilize a warm-up scheme at the first epoch, which is important for the tuning of prompts. For all the other baselines, we strictly follow the configurations of their original papers.

To verify the effectiveness of our proposed method, we explore the improvement of integrating DIP into a lightweight prompt tuning method CoOp and a heavy prompt tuning method MaPLe separately.

For DIP+CoOp and DIP+MaPLe, we conduct a grid search to find the optimal hyper-parameters based on the configuration of CoOp and MaPLe. In the main text, we set the ID order $k = 1$ in DIP for all the experiments unless specially mentioned. For DIP+MaPLe, we also decompose its weights of the projection layer that projects text prompts to generate image prompts with ID order $k_{proj} = 64$. 9 layers are modified in MaPLe by default.

Table 9: Base-to-new generaliation performances based on SLIP Mu et al. (2022).

|  | base | new | H |
|---|---|---|---|
| CoOp | **68.45** | 42.77 | 52.64 |
| DIP+CoOp | 62.53 | **47.78** | **54.17** |

Table 10: Results of different combinations of learning rates and weight decays under 16-shot learning setting on ImageNet

| Acc \ wd / lr | 1e-4 | 5e-4 | 1e-3 |
|---|---|---|---|
| 1e-3 | 70.73 | 70.67 | 70.78 |
| 2e-3 | 70.80 | 70.83 | 70.77 |
| 3e-3 | 70.83 | 70.83 | 70.81 |

## C COMPETITORS

1. **CLIP** Radford et al. (2021): CLIP is a strong baseline vision-language model that is pre-trained on a large number of image-text pairs from the web by learning a contrastive objective. CLIP enables strong zero-shot adaptation ability on various downstream tasks by using fixed text prompts, *i.e. a photo of a*.

2. **CoOp** Zhou et al. (2022a): CoOp replaces the fixed text prompts in CLIP with tunable text prompts to improve the adaptation ability of the vision-language model. CoOp shows excellent performance in few-shot situations.

3. **CoCoOp** Zhou et al. (2022b): CoCoOp replaces the isolated tunable text prompts in CoOp with conditional text prompts, which receive extra gradients from the image features besides text features. CoCoOp largely improves the generalization ability of the vision-language model, getting good results on base-to-new generalization and domain adaptation.

4. **CLIP-Adapter** Gao et al. (2021): CLIP-Adapter adopts the thoughts of classic Adapter Houlsby et al. (2019) to use serial linear layers and activation functions to adapt for downstream tasks. It is simple yet effective in few-shot learning.

5. **LoRA** Hu et al. (2021): LoRA adopts low-rank decomposition for weights in FC layers. It is efficient and earns good results in NLP field.

6. **MaPLe** Khattak et al. (2023a): MaPLe simultaneously adds prompts to the image encoder and text encoder of CLIP. To trigger more information exchange between the image side and text side, MaPLe generate image prompts from the projection of text prompts. Though effective, such design brings quite heavy cost.

7. **ProGrad** Zhu et al. (2023): ProGrad only updates the text and image prompts whose gradient are aligned (or non-conflicting) to the general knowledge, which is represented as the optimization direction offered by the pre-defined prompt predictions. Such regularization helps it finish good adaptation and generalization.

Table 11: Deep prompts in different depths for DIP+CoOp.

| depth | base | new | H |
|---|---|---|---|
| 1 | 76.37 | 74.69 | 75.52 |
| 2 | 77.81 | 73.87 | 75.79 |
| 3 | 79.55 | 72.72 | 75.98 |
| 4 | 80.07 | 72.69 | 76.20 |
| 5 | 80.24 | 73.37 | 76.65 |
| 6 | 80.64 | 73.39 | 76.85 |

Table 12: Training, storage, and inference efficiencies.

| | #params | Training throughput | Inference throughput |
|---|---|---|---|
| CoOp | 2.1K | **93** image/s | **738** images/s |
| CoCoOp | 35.4K | 5 images/s | 13 images/s |
| ProGrad | 8.2K | 56 images/s | 732 images/s |
| DIP+CoOp | **0.5K** | 91 images/s | **738** images/s |

## D  EXPERIMENTS ON A DIFFERENT VISION-LANGUAGE ARCHITECTURE

In this paragraph, we show the results of DIP on another vision-language architecture, SLIP Mu et al. (2022), besides CLIP. Seen from Tab. 9, DIP+CoOp earns much higher new accuracy and harmonic mean accuracy than the original CoOp.

## E  EFFECT OF DEEP PROMPTS

In this paragraph, we extend the shallow prompts in DIP+CoOp to the deep prompts. We record the accuracy change as we increase the layers including tunable prompts, following the last equation in Section 3.1 in the main text. Results are shown in Tab. 11. As the depth increases, the base accuracy keeps increasing while the new accuracy first decreases and then increases. Overall, more depth generally leads to higher harmonic mean accuracy. Therefore, it is possible to further improve the performance of our method by increasing the prompt depth.

## F  EFFECT OF DIFFERENT LEARNING RATES AND WEIGHT DECAYS

In this paragraph, we investigate the effect of normal hyper-parameters learning rate and weight decay. We wonder if tuning them carefully would lead to significant improvement. Seen from Tab. 10, the performance of DIP stays stable whatever the learning rate and weight decay vary. The conclusion here is that our method DIP is robust for learning rate and weight decay.

## G  EFFICIENCY COMPARISON

In this paragraph, we will give a comprehensive analysis of all the training, storage, and inference efficiencies for DIP and several existing methods. Since the parameter scale of CLIP-Adapter Gao et al. (2021) is significantly larger than others, we do not contain CLIP-Adapter into comparison.

For a fair comparison, we do all the speed tests on the same GPU. Results are shown in Tab. 12. Our proposed DIP shares nearly the same fastest training and inference speeds with the simplest method CoOp, and more importantly, DIP merely uses 0.5K parameters, which is a lot more storage-efficient than other methods. Specially, compared with classic method CoCoOp, DIP enjoys >**18x** training speed, >**56x** inference speed and <**70x** storage usage. Besides complex structures, the huge gap in the inference speed is partly owing to the huge memory cost of CoCoOp, which forces us to adopt a smaller batch size than other methods for CoCoOp. Moreover, compared with the latest method ProGrad, DIP enjoys >**1.6x** training speed, comparable inference speed, and <**60x** storage usage, which adequately demonstrates the super efficiency of DIP.

