# OpenReview forum: "Towards Efficient Vision-Language Tuning: More Information Density, More Generalizability"
_ICLR.cc/2025/Conference — ICLR 2025 Conference Withdrawn Submission_

### Official Review · Reviewer_UxJR · 2024-10-30

**Soundness:** 2
**Presentation:** 2
**Contribution:** 2
**Rating:** 5
**Confidence:** 5

**Summary:**

The paper introduces "Information Density" (ID) to measure matrix affiliation with specific feature spaces, hypothesizing that higher ID correlates with robustness and stability. It proposes the Dense Information Prompt (DIP) to boost generalization by increasing ID, with several effective algorithms presented to achieve this.

**Strengths:**

**[New insight]** This work investigates the correlation between information density and generalization in the VLMs domain, which is inspiring.

**Weaknesses:**

**[Dataset shown in Figure 1]** Illustrating the correlation between Information Density and Accuracy is significant, which motivates the method design. Hence, instead of testing on small datasets, it would be nice to experiment on a more representative dataset, e.g., ImageNet. This is because ImageNet is a well-known dataset with more diverse classes, making it a suitable dataset for this investigation. Additionally, it would be better to show results with more baselines, e.g., CoCoOp [1], MaPLe [2] and PromptSRC [3], to convince the readers. This can provide more comprehensive understanding whether this observation suits different types of prompt tuning methods.

[1] Conditional prompt learning for vision-language models. CVPR 2022.

[2] Maple: Multi-modal prompt learning. CVPR 2023.

[3] Self-regulating prompts: Foundational model adaptation without
Forgetting. ICCV 2023.

**[Need evidences]** In section 3.2, to explain information density, the authors draw an analogy using images from nature. It would be more convincing if some previous works containing this philosophy can be provided. For examples, the authors can provide some examples of related work that use similar analogies or concepts to explain information density or related ideas in machine learning.

**[Need more explanation]**
- In section 4.1.1, for Direct Optimization, it needs more details about how the method optimizes IDk to update the original P, i.e.,  a step-by-step explanation of the optimization process for IDk.
- In Table 1, it should introduce the baseline method and the dataset evaluated on. Please add these information in the caption.

**[Wrong equation]** In line 267, the last equation seems wrong, i.e., the second item differs from the one given in line 263.

**[Unclear illustration]** In Figure 2 (a), it should show the difference between these two methods, e.g., DIP has SVD and new loss function. It would be nice to add labels or annotations to highlight these key differences.

**[Experiments]**
- In Table 2, the results of DIP + PromptSRC [1] is missed. It would evaluate the method with diverse types of prompt tuning.
- In Table 3 and 4, some baseline methods, e.g., MaPLe [2], PromptSRC [1] and DePT [3], are missed.
- Training time comparison should be given. This would give a more comprehensive understanding of the effectiveness of the method in various aspects

[1] Self-regulating prompts: Foundational model adaptation without Forgetting. ICCV 2023.

[2] Maple: Multi-modal prompt learning. CVPR 2023.

[3] Dept: Decoupled prompt tuning. CVPR 2024.

**Questions:**

Is the number of trainable parameters same for three DIP variants?

---

### Official Review · Reviewer_fFnU · 2024-11-02

**Soundness:** 2
**Presentation:** 2
**Contribution:** 1
**Rating:** 3
**Confidence:** 5

**Summary:**

This paper tries to leverage the Information Density idea to promote prompt tuning for fine-tuning CLIP to be evaluated on downstream tasks. Specifically, the authors implement such  “Information Density” (ID)  via SVD decomposition, positional penalty, and LoRA to enhance the training, simultaneously DropOut is used to regularize the model not to overfit. Regarding the experiments, the authors present improvements when applied to commonly prompt-based methods, like CoOp, ProGrad, MaPLe, and DePT.

**Strengths:**

1. The experimental results are improved when employed in several prompt-based methods.

2. Attempting to explore the Information Density, being similar to information theory, is kind of interesting.

**Weaknesses:**

1. This paper's motivation apparently lacks logical connections between information density and prompt tuning, which shows confused connections between these two parts. Why can applying the SVD decomposition to obtain the maximal values convey information density, or will the model get into a trivial solution that only these K values works while all the others get zero values? If the authors just fix the maximal K values, essentially, this is truncated SVD and can be analogous to LoRA with k ranks.

2. Regarding the optimization objective L_DO, it is quite weird that the CE term is optimized to be minimized while controlling the λ_DO to be positive, thus the λ_DO * ID_k  is still positive. This results in a decrease in the optimization procedure of cross-entropy loss for this specific downstream task, which can be validated in Table 1 that the base accuracies are inferior to the baseline and are opposite to our fine-tuning target.

3. Based on Weakness 2, I think this sounds like a tricky implementation to regularize the model so as not to overfit the base classes too much and allow the model to perform well at the unseen classes. However, the authors show no in-depth for this part. What's worse, this is totally irrelevant to the information density while considering just applying a loss factor lower than 1.0 for the cross-entropy loss. They need to present the information density change after applying such DIP tuning compared with the baseline.

4. It seems the overall method necessitates to ensemble of LoRA DIP and PP, together with the original prompt vector to obtain final improved results, whereas a dropout is applied to the DIP blocks which shows great improvements but it is a widely used trick. This confuses whether the proposed method shows improvements, or just ensemble some tricks to get improved results rather than coming from the initial "information density" idea.

5. Lots of ablative studies are missed in Table 5. When applying 'Initialization', we can see that the new classes' accuracy improves greatly while further applying 'Regularization' shows marginal improvements. How about removing DO, PP, and LRA within DIP to see the effect of each component?

**Questions:**

1. It seems there is confusion to capture how this paper conveys and demonstrates the information density in the prompt tuning. They need to conduct some experiments that information density clearly correlates with generalizability and adaptation. The authors should demonstrate that this differs from the LoRA, as truncated SVD is analogous to LoRA. While from the Figure 1, it is not convincing to deliver that the Spearman correlation coefficient ρ between information density and new classes accuracy is high and that means the information density strongly correlates with the generalizability, and how about the decrease in terms of the base classes.

---

### Official Review · Reviewer_oaZ4 · 2024-11-04

**Soundness:** 3
**Presentation:** 3
**Contribution:** 3
**Rating:** 5
**Confidence:** 4

**Summary:**

The paper proposes a prompt tuning method based on information density that updates only important parameters. This approach is memory-efficient as it reduces the number of parameters and can be easily applied to various prompt tuning methods. Additionally, the authors proposed initialization and regularization techniques when introducing DIP to enhance generalization performance.

**Strengths:**

1. DIP can be easily applied in a plug-and-play manner to existing prompt tuning methods without significant changes to the model architecture.
2. The performance of DIP has been demonstrated on various benchmark datasets.
3. It is efficient by selecting and updating only important parameters compared to conventional prompt tuning methods.

**Weaknesses:**

1. The claim by the authors that “ID and generalizability are correlated” lacks sufficient experimental evidence. Specifically, in Figure 1, analysis was conducted on only 2 out of 11 datasets, making it difficult to confirm whether this claim is limited to certain datasets or is generally applicable. To enhance persuasiveness, showing similar trends for the remaining datasets and demonstrating generalizability across each dataset would be beneficial. Moreover, in Table 2, only average values are reported, so it is unclear if generalizability consistently improves.
2. Does DIP robust to hyperparameters $\lambda_{DO}$ and $\lambda_{PP}$? Analysis of hyperparameter sensitivity is omitted.
3. As shown in the results in Table 8, there are some limitations when applied to visual prompts.

**Questions:**

Is there any performance improvement when applying DIP to visual prompts of MaPLe?

---

### Note · Authors · 2024-11-15

I have read and agree with the venue's withdrawal policy on behalf of myself and my co-authors.